# A Nomogram for Predicting Prostate Cancer with Lymph Node Involvement in Robot-Assisted Radical Prostatectomy Era: A Retrospective Multicenter Cohort Study in Japan (The MSUG94 Group)

**DOI:** 10.3390/diagnostics12102545

**Published:** 2022-10-20

**Authors:** Makoto Kawase, Shin Ebara, Tomoyuki Tatenuma, Takeshi Sasaki, Yoshinori Ikehata, Akinori Nakayama, Masahiro Toide, Tatsuaki Yoneda, Kazushige Sakaguchi, Takuma Ishihara, Jun Teishima, Kazuhide Makiyama, Takahiro Inoue, Hiroshi Kitamura, Kazutaka Saito, Fumitaka Koga, Shinji Urakami, Takuya Koie

**Affiliations:** 1Department of Urology, Gifu University Graduate School of Medicine, Gifu 5011194, Japan; 2Department of Urology, Hiroshima City Hiroshima Citizens Hospital, Hiroshima 7308518, Japan; 3Department of Urology, Yokohama City University, Yokohama 2360004, Japan; 4Department of Nephro-Urologic Surgery and Andrology, Mie University Graduate School of Medicine, Tsu 5148507, Japan; 5Department of Urology, University of Toyama, Toyama 9300194, Japan; 6Department of Urology, Dokkyo Medical University Saitama Medical Center, Koshigaya 3438555, Japan; 7Department of Urology, Tokyo Metropolitan Cancer and Infectious Diseases Center Komagome Hospital, Tokyo 1138677, Japan; 8Department of Urology, Seirei Hamamatsu General Hospital, Hamamatsu 4308558, Japan; 9Department of Urology, Toranomon Hospital, Tokyo 1058470, Japan; 10Innovative and Clinical Research Promotion Center, Gifu University Hospital, Gifu 5011194, Japan; 11Department of Urology, Kobe City Hospital Organization Kobe City Medical Center West Hospital, Kobe 6530013, Japan

**Keywords:** multicenter cohort study, pelvic lymph node dissection, prostate cancer, robot-assisted radical prostatectomy

## Abstract

Background: To create a nomogram for predicting prostate cancer (PCa) with lymph node involvement (LNI) in the robot-assisted radical prostatectomy (RARP) era. Methods: A retrospective multicenter cohort study was conducted on 3195 patients with PCa who underwent RARP at nine institutions in Japan between September 2012 and August 2021. A multivariable logistic regression model was used to identify factors strongly associated with LNI. The Bootstrap-area under the curve (AUC) was calculated to assess the internal validity of the prediction model. Results: A total of 1855 patients were enrolled in this study. Overall, 93 patients (5.0%) had LNI. On multivariable analyses, initial prostate-specific antigen, number of cancer-positive and-negative biopsy cores, biopsy Gleason grade, and clinical T stage were independent predictors of PCa with LNI. The nomogram predicting PCa with LNI has been demonstrated (AUC 84%). Using a nomogram cut-off of 6%, 492 of 1855 patients (26.5%) would avoid unnecessary pelvic lymph node dissection, and PCa with LNI would be missed in two patients (0.1%). The sensitivity, specificity, and negative predictive values associated with a cutoff of 6% were 74%, 80%, and 99.6%, respectively. Conclusions: We developed a clinically applicable nomogram for predicting the probability of patients with PCa with LNI.

## 1. Introduction

Prostate cancer (PCa) is the second most common cancer in men, accounting for 7% of newly diagnosed male cancers worldwide [1]. For many PCa patients, the disease is a slow-growing and often indolent tumor that requires tailor-made treatment for each individual patient [1]. Life expectancy for men with localized PCa can be as high as 99% at 10 years if diagnosed an early stage of PCa [2]. Approximately 80% of PCa patients are diagnosed with organ-confined disease, 15% with locoregional metastases and 5% with distant metastases [3]. Aggressive PCa that recurs early, independent of definitive treatment for the prostate, is sometimes experienced as unresponsive to standard therapy [1,2]. Although the treatment options for PCa include radical prostatectomy (RP), radiation therapy, and active surveillance therapy based on the pathology of the prostate biopsy specimen, the morphologic aspects of PCa play an important role in the management and prognosis of PCa patients. [2]. This includes parameters such as extracapsular extension, seminal vesicle invasion, perineural invasion, and lymphatic invasion, as well as tumor quantification [2].

RP is one of the treatment options for localized or some advanced PCa according to several guidelines [4,5]. Robot-assisted RP (RARP) is a minimally invasive surgery that has several potential advantages, including reduced bleeding and transfusion rates, postoperative pain, and hospital stay [6]. Pelvic lymph node dissection (PLND) is a golden standard for detecting occult lymph node involvement (LNI) and confirming the accurate staging of high-risk PCa [7]. However, its therapeutic role, indication, and the extent of PLND remain controversial [8]. If LNI can be predicted from preoperative parameters, unnecessary PLND can be avoided. The European Association of Urology (EAU) and the National Comprehensive Cancer Network (NCCN) guidelines recommend the use of nomograms to guide patient selection for extended PLND (ePLND) [4,9]. At present, the Partin table, the Memorial Sloan Kettering Cancer Center (MSKCC), and the Briganti nomogram are all recommended nomograms [10,11,12]. In the NCCN guidelines, PLND can be excluded in patients with <2% predicted probability of LNI by nomogram, although some patients with LNI will be missed. Briganti et al. reported that using a 5% nomogram cut-off, 385 of 588 patients (65.5%) would be spared ePLND and LNI would be missed in only six patients (1.5%) [11]. Gandaglia et al. conducted an external validation of the 2019 Briganti nomogram for estimating LNI risk in 487 patients, which had an area under the curve (AUC) of 79%. They reported that, for a cut-off of 7%, 273 (56%), ePLND would be spared and only 2.6% of LNI would be missed [13]. In a validation study using a contemporary cohort of patients with PCa, the 2012 Briganti and 2018 MSKCC nomograms were identified as the most accurate prediction tools available, with reported AUC of 0.76 and 0.75, respectively [14].

However, these nomograms had complex features that might not be available in clinical practice and included patients who underwent retropubic and laparoscopic RP. This study aimed to create a clinically applicable nomogram for use in the RARP era.

## 2. Materials and Methods

### 2.1. Patient Population

This study was approved by the Institutional Review Board of Gifu University (approval number: 2021-A050) and the institutional review boards of the participating institutions. The requirement for informed consent from the patients was waived because this was a retrospective study. Based on the provisions of the ethics committee and ethics guidelines in Japan, written consent was not required. This is because the results of retrospective and observational studies using materials such as existing documentation had already been disclosed to the public. The details of the study can be found at https://www.med.gifu-u.ac.jp/visitors/disclosure/docs/2021-B039.pdf (accessed on 30 September 2022).

We conducted a retrospective multicenter cohort study of 3195 patients with PCa who underwent RARP at nine institutions in Japan between September 2012 and August 2021 (the MSUG94 cohort). Preoperative patient characteristics were as follows: age, height, weight, serum prostate-specific antigen (PSA) level, prostate volume (PV), clinical stage, biopsy Gleason grade (GG), number of cancer-positive and-negative biopsy cores, NCCN risk stratification [9], performance status according to the Eastern Cooperative Oncology Group (ECOG-PS), and a history of neoadjuvant therapy. The following pathological characteristics were recorded: tumor (T) and node (N) stages of the surgical specimens, GG, status of extraprostatic extension, seminal vesicle invasion, and positive surgical margin (PSM) status. Tumor staging was based on the American Joint Committee on Cancer, eighth edition, Cancer Staging Manual [15].

In this study, the enrolled patients underwent RARP and PLND. The presence or absence of PLND, the extent of PLND, and nerve-sparing approach were determined by the surgeon’s preference or the policy of each institution. The extent of PLND was categorized as follows: limited (including the obturator fossa only); and extended (performed up to the crossing of the common iliac vessel-ureters, with or without the presacral lymph nodes) [16,17].

### 2.2. Pathological Analysis

To evaluate all prostatectomy specimens, we used the whole-mount staining technique and the International Society of Urologic Pathology 2005 guidelines [18]. We truncated the apical section of the prostate perpendicular to the prostatic urethra. The bladder neck margin was coned from the specimen and sectioned perpendicular to this. The remaining prostate tissue was completely sectioned at 3–5 mm intervals along a plane that was perpendicular to the urethral axis.

### 2.3. Statistical Analysis

The primary and secondary endpoints were LNI and the association between LNI and clinical covariates, respectively. We aimed to create a nomogram for predicting PCa with LNI. Patient characteristics of the MSUG94 cohort are described as median and interquartile range for continuous variables and count and proportion for categorical variables. A multivariable logistic regression model was used to identify the factors that were strongly associated with LNI. A prediction model consisting of factors strongly associated with LNI was constructed, and the receiver operating characteristic (ROC) curve and AUC were calculated using the predicted values from the multivariate logistic regression model. The Bootstrap-AUC was calculated using 10,000 bootstrap sets to assess the internal validity of the prediction model. The multiple imputation method was used to assign values to the missing values of the factors in the regression modeling. A two-sided *p*-value of <0.05 was considered significant. All statistical analyses were performed using R 4.1.1 (the R Foundation for Statistical Computing).

## 3. Results

### 3.1. Patient Characteristics

Table 1 shows the demographics of the enrolled patients who underwent RARP and PLND. Patients who underwent preoperative treatment, were diagnosed by transurethral resection of the prostate, had more than 20 negative biopsy cores after prostate biopsy, or were not evaluated for pathological N stage were excluded from this study. Consequently, 1855 patients fulfilled the criteria of the current study.

### 3.2. Development of a Clinically Applicable Nomogram Predicting PCa with Lymph Node Involvement

Overall, 93 patients (5.0%) had LNI. On multivariable analyses, the initial PSA, the number of cancer-positive and-negative biopsy cores, the biopsy GG, and the clinical T stage were all independent predictors of PCa with LNI (Table 2).

Figure 1A graphically displays the multivariable effect of the predictor variables on the risk of LNI in the form of a nomogram. Figure 1B indicates the validated ROC curve of the MSUG94 cohort with an AUC of 84%.

Table 3 shows the predictive accuracy and errors associated with the nomograms. This predicted that patients would have a low risk of LNI. The predicted probabilities of PCa with LNI were categorized. Using a nomogram cut-off of 6%, 492 of 1855 patients (26.5%) would avoid unnecessary PLND, and PCa with LNI would be missed in two patients (0.1%). The sensitivity, specificity, and negative predictive value associated with the 6% cutoff were 74%, 80%, and 99.6%, respectively.

## 4. Discussion

We created a clinically applicable nomogram to predict PCa with LNI in accordance with the RARP era. The role of the nomogram would be to provide a more accurate preoperative diagnosis of LNI and avoid unnecessary PLND. Extended pelvic lymphadenectomy has been established as the gold standard for the detection of LNI [7]. The NCCN guidelines recommend PLND for patients with an intermediate or high risk of PCa and LNI of ≥2% regarding nomograms [9]. However, the oncologic value of PLND for patients with Pca is still controversial [19]. Yang et al. indicated that, in patients with a probability of LNI ≥ 37% by Briganti 2012 nomogram, PLND would improve overall survival rate [20]. Lestingi et al. reported that PLND may have potential oncological benefits for patients who have been diagnosed with GG 3–5 [19]. If PLND is unlikely to benefit the patients with PCa, it may not be performed at RARP. In this study, using a nomogram cut-off of 10%, 767 patients (41.3%) would avoid unnecessary PLND and PCa with LNI would be missed in 8 patients (0.4%); using a nomogram cut-off of 20%, 1227 patients (66.1%) would avoid unnecessary PLND and PCa with LNI would be missed in 23 patients (1.2%). Compared with Briganti and MSKCC nomograms, our nomogram indicated a high AUC (84%) and a high negative predictive value (99.6%).

The Briganti nomograms were the first to update a nomogram predicting the presence of LNI in patients treated with PLND [11]. The nomogram contained PSA, clinical T stage at magnetic resonance imaging (MRI), primary Gleason score (GS), secondary GS, and percentage of positive cores. The AUC of Briganti nomogram 2012 was 79.5% [11]. The advantages of Briganti nomogram 2012 were composed of factors that were easy to use. Briganti et al. revised their nomograms in 2017 and 2019, respectively. The Briganti nomogram 2017 consisted of five factors: biopsy GG, clinical T stage, preoperative PSA, percentage of positive cores with highest-grade disease, and percentage of positive cores with lower-grade disease. The AUC of Briganti nomogram 2017 was 90.8% [21]. The new nomogram of Briganti included complex factors such as PSA, clinical T stage at MRI, GG at MRI-targeted biopsy, maximum diameter of the index lesion at MRI, and percentage of cores with clinically significant PCa at systematic biopsy. The AUC of the Briganti nomogram 2019 was 86.0% [22]. It was somewhat cumbersome to clinically apply this to all patients with Pca, despite the undoubtedly useful nomogram. Our data consisted of only simple factors, but comparable AUCs were obtained because of the large number of cases compared with the Briganti nomogram 2019, which included 497 patients. The MSKCC nomogram noted the AUC at 84.6% [12]. The factors are similar to our nomogram. The advantage of the MSKCC nomogram is that it provides multiple pieces of information simultaneously, for example, extracapsular extension, seminal vesicle invasion, 15-year prostate cancer-specific survival, and progression-free probability after RP. In another nomogram, Nave et al. describes the mathematical model that combined the singular perturbed vector field and the method of directly defining the inverse mapping to effect immunotherapy and androgen deprivation therapy for advanced PCa [23,24]. This nomogram contains a system of nonlinear ordinary differential equations of the first order and described the interaction between androgen-dependent cancer cells, androgen-independent cancer cells, activated T cells, cytokine concentration, androgen concentration and dendritic cell number [24]. Additionally, this nomogram showed that the optimal interaction of immunotherapy was the interaction described by the appropriate coefficients and the fast direction of the system [24]. Our nomogram is better than others because it is based on recent RARP patients and uses intuitive factors. We were able to produce a clinically applicable nomogram for predicting PCa with LNI. The cut-off for recommending LND was 2% in the NCCN guidelines. They applied the 2% threshold based on the rationale that 47.7% of PLNDs could be avoided at the expense of missing 12.1% of patients with LNI [25]. To address this cutoff issue, we tested alternative thresholds of 10%, 20%, and 30%. These thresholds resulted in proportions of 41.3%, 66.1%, and 79.0%, respectively, in which PLND could be avoided. However, this would occur at the expense of missing 0.4%, 1.2%, and 2.1% of LNI patients. Using a nomogram cut-off of 30%, the cost of missing 39 patients with LNI (2.1%) could be acceptable. However, it should be noted that our cohort included only 93 patients with LNI. The 39 patients represent 41.9% of all LNI patients. In the current situation, where PLND has no apparent benefit in terms of overall survival rate, we believe that the cutoff should be approximately 10%.

To date, ePLND is the most accurate technique for detecting occult LNI in PCa patients [1,2,4,5]. However, Claps et al. reported that free-Indocyanine Green (F-ICG) could accurately assess pathologic lymph node (LN) status and based on its high negative predictive value [26] Therefore, it is safe to avoid PLND for most patients if pN0 on F-ICG staining [26]. In addition, because F-ICG avoids unnecessary PLND, liquid biopsy has recently gained attention as a biomarker to inform clinical decision-making in PCa [27]. Recently, biomarkers have been explored for various urologic cancers, including prostate cancer [28,29,30,31,32,33,34]. Liquid biopsy is an emerging biomarker for clinical decision making in PCa [35]. In patients with metastatic PCa, immunoinflammatory values were reported to be an independent prognostic factor for overall survival [36]. The most clinically studied areas of liquid biopsy are circulating tumor cells and circulating tumor DNA, both of which are considered useful prognostic markers for metastatic prostate cancer [36]. In this study, inflammatory markers such as neutrophil-to-lymphocyte ratio, platelet-to-lymphocyte ratio, and systemic immune-inflammatory index were not correlated with LNI in PCa patients undergoing RARP. In the near future, PCa with aggressive features and potential lymph node metastasis may be predicted by liquid biopsy.

To date, no nomograms have predicted PCa with LNI in all patients undergoing RARP. Our strength was the use of a relatively large cohort and that all patients underwent RARP. Our nomogram consisted of accessibility to five factors and allowed for easy calculation of the probability of LNI. Despite its strengths, this study has several limitations. First, this study was retrospective in the nature of multicenter data, which may have conferred susceptibility to potential selection bias as diagnostic and surgical approaches varied between the included institutions. Second, the presence or absence of PLND and the range of PLND were determined by the surgeon’s preference or the policy of each institution. In this study, 93 patients (5.0%) received PLND and only 17.3% of eligible patients underwent ePLND, with a median of 7 LNs. Third, the prostate biopsy core, prostatectomy, and PLND specimens were not re-evaluated by a single pathologist. However, Ghadjar et al. reported that the results of the central pathology analyses showed concordance between the central and local pathology reviews [37]. Our data lacked variant histological type PCa; Humphrey et al. reviewed the pathologic classification of urologic malignancies in 2016 and found that variant histological types may harbor different biological behavior [38]. Fourth, the diagnosis of clinical stage on MRI and digital rectal examination differs among surgeons. The T-stage using MRI for nomogram resulted in higher AUC and a higher net benefit compared with the use of digital rectal examination in both the MSKCC 2018 and Briganti 2012 nomograms [39]. However, in this study, no data were collected on whether the enrolled patients underwent evaluation using magnetic resonance imaging before prostate biopsy. Finally, we assessed the internal validity of the prediction model using bootstrap-AUC, but external validation was not performed. Further research is warranted to identify more accurate decision-making tools that will help in reducing unnecessary invasion, accurately identifying relevant patients, and minimizing the number of LNI that are missed.

There are several limitations to this study. First, this was a retrospective, multicenter cohort study and might results in bias. Second, we acknowledge that pathology review, including biopsy and pathology GS, is not centralized in this study. Third, the relatively short follow-up period may be insufficient to accurately identify predictors of BCR after RARP. Finally, the number for dissecting lymph node counts and the quality of PLND may not be consistent because the extent of lymph node dissection, which varied according to surgeon preference or the policy of each institution.

## 5. Conclusions

The nomograms available at this time have complex features that may make them difficult to use in clinical practice, and they are intended for patients who have undergone open and laparoscopic RP. We developed and internally validated a clinically applicable nomogram to predict the probability of patients with PCa with LNI undergoing RARP. It is based on available clinical parameters such as PSA, clinical T stage, biopsy GG, and the number of positive and negative cores. Using a 10% cut-off, 41.3% of patients could avoid PLND and would be missed by only 0.4%. Thus, in the current absence of prospective data supporting the role of PLND in oncologic outcomes of PCa, the avoidance of PLND may not be problematic in patients with a nomogram-based LNI risk of less than 10%.

## Figures and Tables

**Figure 1 diagnostics-12-02545-f001:**
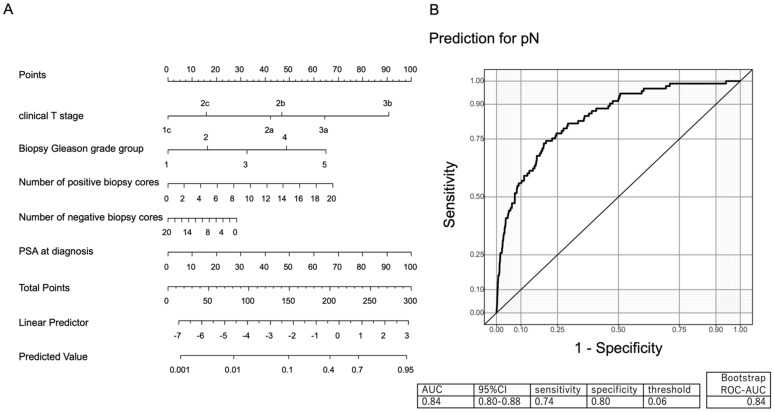
(**A**) Our nomogram predicting the probability of prostate cancer (PCa) with lymph node involvement (LNI) in patients undergoing robot-assisted radical prostatectomy and pelvic lymph node dissection using clinical T stage, biopsy Gleason grade, number of positive biopsy cores, number of negative biopsy cores, and prostate-specific antigen at diagnosis. (**B**) Receiver operating characteristic curve of predictive accuracy for PCa with LNI.

**Table 1 diagnostics-12-02545-t001:** Patient demographics.

Variables	
Age (year, median, IQR)	68 (64–72)
Body mass index (median, IQR)	23.7 (21.8–25.7)
ECOG Performance Status (number, %)	
0	1800 (97.0)
1	51 (2.7)
2	4 (0.2)
Initial PSA (ng/mL, median, IQR)	8.1 (5.8–12.2)
Prostate volume (cc, median, IQR)	30.0 (22.8–39.4)
Number of biopsy cores (median, IQR)	10 (10–13)
Number of positive cores (median, IQR)	4 (2–5)
Number of negative cores (median, IQR)	8 (6–10)
Biopsy Gleason Grade (number, %) 1 2 3 4 5	182 (9.8) 572 (30.8) 442 (23.8) 488 (26.3) 171 (9.2)
Clinical T stage (number, %)	
T1c	326 (17.6)
T2a	770 (41.5)
T2b	198 (10.7)
T2c	364 (19.6)
T3a	179 (9.7)
T3b	18 (1.0)
Hemoglobin (g/dL, median, IQR)	14.5 (13.8–15.2)
C-reactive protein (mg/dL, median, IQR)	0.06 (0.03–0.11)
NLR (median, IQR)	2.01 (1.52–2.71)
PLR (median, IQR)	125 (97.1–163)
SII (median, IQR)	417 (294–579)
Pelvic lymphadenectomy (number, %) Limited Standard Extended	1312 (70.7) 214 (11.5) 320 (17.3)
Removed and examined lymph nodes count (median, IQR)	7 (4–11)
Pathological Gleason Grade (number, %) 1 2 3 4 5	182 (9.8) 572 (30.8) 442 (23.8) 488 (26.3) 171 (9.2)
Pathological T stage (number, %) T2 T3 T4	1206 (65.0) 641 (34.6) 7 (0.4)
Positive surgical margin (number, %)	632 (34.1)
Lymph node involvement (number, %)	93 (5.0)

IQR: interquartile range; ECOG: The Eastern Cooperative Oncology Group; PSA: Prostate-specific antigen; NLR: neutrophil-to-lymphocyte ratio; PLR: platelet-to-lymphocyte ratio; SII: systemic immune-inflammation index.

**Table 2 diagnostics-12-02545-t002:** Multivariable logistic regression analysis in the MSUG94 cohort.

Variables	Odds Ratio	95% Confidence Interval	*p* Value
Age	1.02	0.75–1.38	0.910
Body mass index	0.95	0.69–1.30	0.748
ECOG-PS	2.19	0.26–18.52	0.470
Initial PSA	1.25	1.11–1.40	<0.001
Prostate volume	1.08	0.82–1.41	0.58
Number of positive cores	1.43	1.13–1.80	0.003
Number of negative cores	0.80	0.59–1.09	0.156
Biopsy Gleason Grade	3.17	2.02–4.99	<0.001
Clinical T stage 2a ^†^	4.59	1.07–19.6	0.04
Clinical T stage 2b ^†^	5.32	1.14–24.8	0.003
Clinical T stage 2c ^†^	1.75	0.37–8.43	0.483
Clinical T stage 3a ^†^	10.05	2.27–44.47	0.002
Clinical T stage 3b ^†^	24.71	4.06–150.29	<0.001
Hemoglobin	1.03	0.77–1.38	0.853
C-reactive protein	0.98	0.69–1.30	0.748
NLR	0.93	0.59–1.48	0.765
PLR	1.00	0.67–1.50	0.999
SII	0.98	0.67–1.44	0.909

ECOG-PS: The Eastern Cooperative Oncology Group Performance Status; PSA: Prostate-specific antigen; NLR: neutrophil-to-lymphocyte ratio; PLR: platelet-to-lymphocyte ratio; SII: systemic immune-inflammation index. ^†^ Compared to T1c.

**Table 3 diagnostics-12-02545-t003:** Systematic analyses of the nomogram-derived probability cutoffs for prostate cancer with lymph node involvement.

Nomogram Cutoff	Patient below Cutoff with pN1	Patient below Cutoff with pN0	Negative Predictive Value
6%	2	492	99.6%
10%	8	767	99.0%
15%	16	1041	98.5%
20%	23	1227	98.2%
25%	30	1362	97.8%
30%	39	1466	97.4%

## Data Availability

The data presented in this study are available on request from the corresponding author. The data are not publicly available due to privacy and ethical reasons.

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
