# Peer review of "A Nomogram for Predicting Prostate Cancer with Lymph Node Involvement in Robot-Assisted Radical Prostatectomy Era: A Retrospective Multicenter Cohort Study in Japan (The MSUG94 Group)"

_diagnostics, 2022, doi:10.3390/diagnostics12102545_

Round 1
Reviewer 1 Report
The paper is well-conducted and it runs in a way of great interest for the readers as well as the prostate cancer is.
Author Response
17, October, 2022
Dear Editor-in-Chief
the Diagnostics
Dear Editor
Thank you very much for the review of our manuscript titled “A Nomogram for Predicting Prostate Cancer with Lymph Node Involvement in Robot-Assisted Radical Prostatectomy Era: A Retrospective Multicenter Cohort Study in Japan (The MSUG94 Group).”
We sincerely appreciate all valuable comments and suggestions, which helped us to improve the quality of our manuscript. Our responses to the Reviewers’ comments are described below in a point-to-point manner. Appropriate changes, suggested by the Reviewers, have been introduced to the manuscript (track-changes mode in the red color font). Let me emphasize our full readiness to make any further improvements to the manuscript.
We hope that our manuscript will be acceptable for publication in the DIagnostics.
We look forward to hearing from you.
Yours sincerely,
Takuya Koie
Corresponding author
Department of Urology
Gifu University Graduate School of Medicine
1-1 Yanagido, Gifu, Gifu 501-1194, Japan
TEL.: +81-582-30-6338
FAX: +81-582-30-6341
e-mail: goodwin@gifu-u.ac.jp
Responses to the reviewer's comments
We would like to thank the Reviewers for taking the time and effort necessary to review the manuscript. We sincerely appreciate all the valuable comments and suggestions, which helped us to improve the quality of the manuscript.
Response to Reviewer 1
The authors appreciate the reviewer’s comments.

Reviewer 2 Report
The paper is very interesting and has the potential to publish in the above journal after the author address the following major comments:
- The abstract is written very well. And summarized the paper well.
- The introduction section should be extended and includes more relevant research that relates to the current research.
- The methodology of the research is not clear. After the introduction section there are some sentences and after that the results of the research. The authors must explain in detail the process of the research, the methodology, the tools that they used to get their results, etc.
- The conclusion section must be extended.
- The following paper can be added to the current papers:
Nave, O., & Elbaz, M. (2018). Method of directly defining the inverse mapping applied to prostate cancer immunotherapy — Mathematical model. In International Journal of Biomathematics (Vol. 11, Issue 05, p. 1850072). World Scientific Pub Co Pte Lt. https://doi.org/10.1142/s1793524518500729
Rebello, R. J., Oing, C., Knudsen, K. E., Loeb, S., Johnson, D. C., Reiter, R. E., Gillessen, S., Van der Kwast, T., & Bristow, R. G. (2021). Prostate cancer. In Nature Reviews Disease Primers (Vol. 7, Issue 1). Springer Science and Business Media LLC. https://doi.org/10.1038/s41572-020-00243-0
Nave, O., & Elbaz, M. (2018). Combination of singularly perturbed vector field method and method of directly defining the inverse mapping applied to complex ODE system prostate cancer model. In Journal of Biological Dynamics (Vol. 12, Issue 1, pp. 961–986). Informa UK Limited. https://doi.org/10.1080/17513758.2018.1541104
Grignon, D. J. (2018). Prostate cancer reporting and staging: needle biopsy and radical prostatectomy specimens. In Modern Pathology (Vol. 31, Issue S1, pp. 96–109). Springer Science and Business Media LLC. https://doi.org/10.1038/modpathol.2017.167
Author Response
17, October, 2022
Dear Editor-in-Chief
the Diagnostics
Dear Editor
Thank you very much for the review of our manuscript titled “A Nomogram for Predicting Prostate Cancer with Lymph Node Involvement in Robot-Assisted Radical Prostatectomy Era: A Retrospective Multicenter Cohort Study in Japan (The MSUG94 Group).”
We sincerely appreciate all valuable comments and suggestions, which helped us to improve the quality of our manuscript. Our responses to the Reviewers’ comments are described below in a point-to-point manner. Appropriate changes, suggested by the Reviewers, have been introduced to the manuscript (track-changes mode in the red color font). Let me emphasize our full readiness to make any further improvements to the manuscript.
We hope that our manuscript will be acceptable for publication in the DIagnostics.
We look forward to hearing from you.
Yours sincerely,
Takuya Koie
Corresponding author
Department of Urology
Gifu University Graduate School of Medicine
1-1 Yanagido, Gifu, Gifu 501-1194, Japan
TEL.: +81-582-30-6338
FAX: +81-582-30-6341
e-mail: goodwin@gifu-u.ac.jp
Responses to the reviewer's comments
We would like to thank the Reviewers for taking the time and effort necessary to review the manuscript. We sincerely appreciate all the valuable comments and suggestions, which helped us to improve the quality of the manuscript.
Response to Reviewer 2
The authors appreciate the reviewer’s comments. The authors’ point-by-point responses to the comments are given below.
1) The abstract is written very well. And summarized the paper well.
Response:
Thank you for your kind comments.
2) The introduction section should be extended and includes more relevant research that relates to the current research.
Response:
The authors have added the following sentence on line 51:
Prostate cancer (PCa) is the second most common cancer in men, accounting for 7% of newly diagnosed male cancers worldwide [1]. For many PCa patients, the disease is a slow-growing and often indolent tumor that requires tailor-made treatment for each individual patient [1]. Life expectancy for men with localized PCa can be as high as 99% at 10 years if diagnosed an early stage of PCa [2]. Approximately 80% of PCa patients are diagnosed with organ-confined disease, 15% with locoregional metastases and 5% with distant metastases [3]. Aggressive PCa that recurs early, independent of definitive treatment for the prostate, is sometimes experienced as unresponsive to standard therapy [1,2]. Although The treatment options for PCa include radical prostatectomy (RP), radiation therapy, and active surveillance therapy based on the pathology of the prostate biopsy specimen, the morphologic aspects of PCa play an important role in the management and prognosis of PCa patients. [2]. This includes parameters such as extracapsular extension, seminal vesicle invasion, perineural invasion, and lymphatic invasion as well as tumor quantification [2].
3) The methodology of the research is not clear. After the introduction section there are some sentences and after that the results of the research. The authors must explain in detail the process of the research, the methodology, the tools that they used to get their results, etc.
Response:
The authors have added the following parts on line 112:
In this study, the enrolled patients underwent RARP and PLND.
The authors have revised the following sentences on line 126:
The primary and secondary endpoints were LNI and the association between LNI and clinical covariates, respectively. We aimed to create a nomogram for predicting PCa with LNI.
On line 131, The authors have already described the methodology of this study as follows:
A prediction model consisting of factors strongly associated with LNI was constructed, and the receiver operating characteristic (ROC) curve and AUC were calculated using the predicted values from the multivariate logistic regression model. The Bootstrap-AUC was calculated using 10,000 bootstrap sets to assess the internal validity of the prediction model. The multiple imputation method was used to assign values to the missing values of the factors in the regression modeling.
4) The conclusion section must be extended.
Response:
The authors have revised the following sentences on line 281:
The nomograms available at this time have complex features that may make them difficult to use in clinical practice, and they are intended for patients who have undergone open and laparoscopic RP. We developed and internally validated a clinically applicable nomogram to predict the probability of patients with PCa with LNI undergoing RARP.
The authors have revised the following sentences on line 287:
Therefore, an accurate diagnosis using nomograms for PCa with LNI could avoid unnecessary PLND. Thus, in the current absence of prospective data supporting the role of PLND in oncologic outcomes of PCa, avoidance of PLND may not be problematic in patients with a nomogram-based LNI risk of less than 10%.
5) The following paper can be added to the current papers:
Response:
The authors have added the following paper as ref.23 and 24:
Nave, O., & Elbaz, M. (2018). Method of directly defining the inverse mapping applied to prostate cancer immunotherapy — Mathematical model. In International Journal of Biomathematics (Vol. 11, Issue 05, p. 1850072). World Scientific Pub Co Pte Lt. https://doi.org/10.1142/s1793524518500729
Nave, O., & Elbaz, M. (2018). Combination of singularly perturbed vector field method and method of directly defining the inverse mapping applied to complex ODE system prostate cancer model. In Journal of Biological Dynamics (Vol. 12, Issue 1, pp. 961–986). Informa UK Limited. https://doi.org/10.1080/17513758.2018.1541104
The authors have added the following paper as ref. 1:
Rebello, R. J., Oing, C., Knudsen, K. E., Loeb, S., Johnson, D. C., Reiter, R. E., Gillessen, S., Van der Kwast, T., & Bristow, R. G. (2021). Prostate cancer. In Nature Reviews Disease Primers (Vol. 7, Issue 1). Springer Science and Business Media LLC. https://doi.org/10.1038/s41572-020-00243-0
The authors have added the following paper as ref. 22:
Grignon, D. J. (2018). Prostate cancer reporting and staging: needle biopsy and radical prostatectomy specimens. In Modern Pathology (Vol. 31, Issue S1, pp. 96–109). Springer Science and Business Media LLC. https://doi.org/10.1038/modpathol.2017.167
The authors have added the following paper as ref. 3:
- Corn, P.G. The tumor microenvironment in prostate cancer: elucidating
molecular pathways for therapy development. Cancer Manag Res. 2012, 4, 183-
193.

Reviewer 3 Report
In this manuscript the authors created a nomogram predicting LNI in the RARP era.
Only 93 (5.0%) patients with LNI were found at time of pathological examination. Moreover, only 17.3% of the included patients undernwent PLND with an extended template. This represent the major limitation of the study because only a median of 7 lymph nodes were per patient removed.
Nomograms are overadopted nowdays and the authors should improved the Discussion paragraph with a paragraph including novel technologies that could be applied to target the lymph node dissection in prostate cancer patients undergoing radical surgery. For example the concept of sentinel node biopsy is an attractive tool to maximize the yield of pathology-specific lymph nodes as recently highlighted (doi: 10.1016/j.urolonc.2022.08.005, doi: 10.1111/iju.14513). Beyond nomograms novel technologies could drive the management of these patients.
As per above, the authors should consider to add an additive paragraph of novel molecular biomarkers in predicting biological aggressiveness among urological malignancies. These emerging tools can potentially drive the prognosis and tailor the treatment. Please consider to such references about prostate cancer itself (Actas Urol Esp (Engl Ed). 2020 Apr;44(3):139-147. doi: 10.1016/j.acuro.2019.08.007), (Eur Urol. 2021 Jun;79(6):762-771. doi: 10.1016/j.eururo.2020.12.037), (Tumour Biol. 2022;44(1):107-127. doi: 10.3233/TUB-211568.), (Prostate. 2022 Nov;82(15):1456-1461. doi: 10.1002/pros.24419. Epub 2022 Jul 28.), (N Engl J Med. 2015 Oct 29;373(18):1697-708) (Cancers (Basel). 2021 Sep 8;13(18):4522. doi: 10.3390/cancers13184522), bladder cancer (Eur Urol Oncol. 2021 May 6;S2588-9311(21)00078-X. doi: 10.1016/j.euo.2021.04.004), (Asian J Urol. 2021 Oct;8(4):376-390. doi: 10.1016/j.ajur.2021.05.001), (Eur Urol Oncol. 2021 Apr;4(2):204-214. doi: 10.1016/j.euo.2020.01.003), (Urol Oncol. 2022 Mar;40(3):110.e1-110.e9. doi: 10.1016/j.urolonc.2021.10.010. Epub 2021 Dec 11. ), and renal cell carcinoma (Lancet Oncol. 2022 May;23(5):612-624. doi: 10.1016/S1470-2045(22)00128-0. Epub 2022 Apr 4.), (J Immunother Cancer. 2022 Mar;10(3):e004316. doi: 10.1136/jitc-2021-004316.).
Further comment about the presence of variant histology prostate cancer. Do you have such information? Humprey et al. in 2016 reviewed the pathological classification of urological malignancies and variant histhology could harbor a different biological behaviour as recently highlighted for the cribriform pattern (doi: 10.1038/s41391-022-00600-y). Please consider to add this limitation.
Author Response
17, October, 2022
Dear Editor-in-Chief
the Diagnostics
Dear Editor
Thank you very much for the review of our manuscript titled “A Nomogram for Predicting Prostate Cancer with Lymph Node Involvement in Robot-Assisted Radical Prostatectomy Era: A Retrospective Multicenter Cohort Study in Japan (The MSUG94 Group).”
We sincerely appreciate all valuable comments and suggestions, which helped us to improve the quality of our manuscript. Our responses to the Reviewers’ comments are described below in a point-to-point manner. Appropriate changes, suggested by the Reviewers, have been introduced to the manuscript (track-changes mode in the red color font). Let me emphasize our full readiness to make any further improvements to the manuscript.
We hope that our manuscript will be acceptable for publication in the DIagnostics.
We look forward to hearing from you.
Yours sincerely,
Takuya Koie
Corresponding author
Department of Urology
Gifu University Graduate School of Medicine
1-1 Yanagido, Gifu, Gifu 501-1194, Japan
TEL.: +81-582-30-6338
FAX: +81-582-30-6341
e-mail: goodwin@gifu-u.ac.jp
Responses to the reviewer's comments
We would like to thank the Reviewers for taking the time and effort necessary to review the manuscript. We sincerely appreciate all the valuable comments and suggestions, which helped us to improve the quality of the manuscript.
Response to Reviewer 3
The authors appreciate the reviewer’s comments. The authors’ point-by-point responses to the comments are given below.
1) Only 93 (5.0%) patients with LNI were found at time of pathological examination. Moreover, only 17.3% of the included patients underwent PLND with an extended template. This represent the major limitation of the study because only a median of 7 lymph nodes were per patient removed.
Response:
The authors have added the following sentence on line 256:
In this study, 93 patients (5.0%) received PLND and only 17.3% of eligible patients underwent ePLND, with a median of 7 LNs.
2) Nomograms are overadopted nowdays and the authors should improved the Discussion paragraph with a paragraph including novel technologies that could be applied to target the lymph node dissection in prostate cancer patients undergoing radical surgery. For example the concept of sentinel node biopsy is an attractive tool to maximize the yield of pathology-specific lymph nodes as recently highlighted (doi: 10.1016/j.urolonc.2022.08.005, doi: 10.1111/iju.14513). Beyond nomograms novel technologies could drive the management of these patients.
Response:
The authors have added the following sentence on line 232:
To date, ePLND is the most accurate technique for detecting occult LNI in PCa patients [1,2,4,5]. However, Claps et al. reported that free-Indocyanine Green (F-ICG) could accurately assess pathologic lymph node (LN) status and based on its high negative predictive value [26] Therefore, it is safe to avoid PLND for most patients if pN0 on F-ICG staining [26]. In addition, because F-ICG avoids unnecessary PLND, liquid biopsy has recently gained attention as a biomarker to inform clinical decision making in PCa [27].
The authors have added the following paper as ref. 26 and 27:
- Francesco Claps, F.; de Pablos-Rodríguez, P.; Gómez-Ferrer, Á.; Mascarós,
J.M.; Marenco, J.; Collado Serra, A.; Ramón-Borja, J.C.; Calatrava Fons, A.;
Trombetta, C.; Rubio-Briones, J.; Ramírez-Backhaus, M. Free-indocyanine
green-guided pelvic lymph node dissection during radical prostatectomy. Urol
Oncol. 2022, in press.
- Claps, F.; Ramírez-Backhaus, M.; Mir Maresma, M.C.; Gómez-Ferrer, Á.;
Mascarós, J.M.; Marenco, J.; Collado Serra, A.; Casanova Ramón-Borja, J.;
Calatrava Fons, A.; Trombetta, C.; Rubio-Briones, J. Indocyanine green guidance
improves the efficiency of extended pelvic lymph node dissection during
laparoscopic radical prostatectomy. Int J Urol. 2021, 28, 566-572.
3) As per above, the authors should consider to add an additive paragraph of novel molecular biomarkers in predicting biological aggressiveness among urological malignancies. These emerging tools can potentially drive the prognosis and tailor the treatment. Please consider to such references about prostate cancer itself (Actas Urol Esp (Engl Ed). 2020 Apr;44(3):139-147. doi: 10.1016/j.acuro.2019.08.007), (Eur Urol. 2021 Jun;79(6):762-771. doi: 10.1016/j.eururo.2020.12.037), (Tumour Biol. 2022;44(1):107-127. doi: 10.3233/TUB-211568.), (Prostate. 2022 Nov;82(15):1456-1461. doi: 10.1002/pros.24419. Epub 2022 Jul 28.), (N Engl J Med. 2015 Oct 29;373(18):1697-708) (Cancers (Basel). 2021 Sep 8;13(18):4522. doi: 10.3390/cancers13184522), bladder cancer (Eur Urol Oncol. 2021 May 6;S2588-9311(21)00078-X. doi: 10.1016/j.euo.2021.04.004), (Asian J Urol. 2021 Oct;8(4):376-390. doi: 10.1016/j.ajur.2021.05.001), (Eur Urol Oncol. 2021 Apr;4(2):204-214. doi: 10.1016/j.euo.2020.01.003), (Urol Oncol. 2022 Mar;40(3):110.e1-110.e9. doi: 10.1016/j.urolonc.2021.10.010. Epub 2021 Dec 11. ), and renal cell carcinoma (Lancet Oncol. 2022 May;23(5):612-624. doi: 10.1016/S1470-2045(22)00128-0. Epub 2022 Apr 4.), (J Immunother Cancer. 2022 Mar;10(3):e004316. doi: 10.1136/jitc-2021-004316.).
Response:
The authors have added the following sentence on line 237:
Recently, biomarkers have been explored for various urologic cancers, including prostate cancer [28-34]. Liquid biopsy is an emerging biomarker for clinical decision making in PCa [35]. In patients with metastatic PCa, immunoinflammatory values were reported to be an independent prognostic factor for overall survival [36]. The most clinically studied areas of liquid biopsy are circulating tumor cells and circulating tumor DNA, both of which are considered useful prognostic markers for metastatic prostate cancer [36]. In this study, inflammatory markers such as neutrophil-to-lymphocyte ratio, plate-let-to-lymphocyte ratio, and systemic immune-inflammatory index were not correlated with LNI in PCa patients undergoing RARP. In the near future, PCa with aggressive fea-tures and potential lymph node metastasis may be predicted by liquid biopsy.
The authors have added the following paper as ref. 28- 34:
- Mir MC, Campi R, Loriot Y, Puente J, Giannarini G, Necchi A, Rouprêt M.
Adjuvant Systemic Therapy for High-risk Muscle-invasive Bladder Cancer After
Radical Cystectomy: Current Options and Future Opportunities. Eur Urol Oncol.
2021, in press.
- Claps, F.; Mir, M.C.; Zargar, H. Molecular markers of systemic therapy
response in urothelial carcinoma. Asian J Urol, 2021, 8, 376-390.
- de Kruijff, I.E.; Beije. N.; Martens, J.W.M.; de Wit. R.; Boormans. J.L.; Sleijfer,
- Liquid Biopsies to Select Patients for Perioperative Chemotherapy in Muscle-
invasive Bladder Cancer: A Systematic Review. Eur Urol Oncol. 2021, 4, 204-
214.
- Vano, Y.A.; Elaidi, R.; Bennamoun, M.; Chevreau, C.; Borchiellini, D.; Pannier,
D.; Maillet, D.; Gross-Goupil, M.; Tournigand, C.; Laguerre, B.; Barthélémy, P.;
Coquan, E.; Gravis, G.; Houede, N.; Cancel, M.; Huillard, O.; Beuzeboc, P.;
Fournier, L.; Méjean, A.; Cathelineau, X.; Doumerc, N.; Paparel, P.; Bernhard,
J.C.; de la Taille, A.; Bensalah, K.; Tricard, T.; Waeckel, T.; Pignot, G.;
Braychenko, E.; Caruso, S.; Sun, C.M.; Verkarre, V.; Lacroix, G.; Moreira, M.;
Meylan, M.; Bougouïn, A.; Phan, L.; Thibault-Carpentier, C.; Zucman-Rossi, J.;
Fridman, W.H.; Sautès-Fridman, C.; Oudard, S. Nivolumab, nivolumab-
ipilimumab, and VEGFR-tyrosine kinase inhibitors as first-line treatment for
metastatic clear-cell renal cell carcinoma (BIONIKK): a biomarker-driven, open-
label, non-comparative, randomised, phase 2 trial. Lancet Oncol. 2022, 23, 612-
624.
- Motzer RJ, Choueiri TK, McDermott DF, Powles T, Vano YA, Gupta S, Yao
J, Han C, Ammar R, Papillon-Cavanagh S, Saggi SS, McHenry MB, Ross-
Macdonald P, Wind-Rotolo M. Biomarker analysis from CheckMate 214:
nivolumab plus ipilimumab versus sunitinib in renal cell carcinoma. J Immunother
Cancer. 2022, 10, e004316.
- Casanova-Salas, I.; Athie, A.; Boutros, PC.; Re, MD.; Miyamoto, DT.; Pienta,
KJ.; Posadas, EM.; Sowalsky, AG.; Stenzl, A.; Wyatt, AW.; et al. Quantitative and
Qualitative Analysis of Blood-based Liquid Biopsies to Inform Clinical Decision-
making in Prostate Cancer. Eur Urol. 2021, 79, 762-771.
- 34. Mertens, L.S.; Claps, F.; Mayr, R.; Bostrom, P.J.; Shariat, S.F.; Zwarthoff,
E.C.; Boormans, J.L.; Abas, C.; van Leenders, G.J.L.H.; Götz. S.; Hippe, K.; Bertz,
S.; Neuzillet, Y.; Sanders, J.; Broeks, A.; Peters, D.; van der Heijden, M.S.; Jewett,
M.A.S.; Stöhr, R.; Zlotta, A.R.; Eckstein, M.; Soorojebally, Y.; van der Schoot,
D.K.E.; Wullich, B.; Burger, M.; Otto, W.; Radvanyi, F.; Sirab, N.; Pouessel, D.;
van der Kwast, T.H.; Hartmann, A.; Lotan, Y.; Allory,Y.; Zuiverloon, T.C.M.; van
Rhijn, B.W.G. Prognostic markers in invasive bladder cancer: FGFR3 mutation
status versus P53 and KI-67 expression: a multi-center, multi-laboratory analysis
in 1058 radical cystectomy patients. Urol Oncol. 2022, 40, 110.e1-100.e9.
4) Further comment about the presence of variant histology prostate cancer. Do you have such information? Humprey et al. in 2016 reviewed the pathological classification of urological malignancies and variant histhology could harbor a different biological behaviour as recently highlighted for the cribriform pattern (doi: 10.1038/s41391-022-00600-y). Please consider to add this limitation.
Response:
The authors have added the following sentence on line 260:
Our data lacked variant histological type PCa; Humphrey et al. reviewed the pathologic classification of urologic malignancies in 2016 and found that variant histological types may harbor different biological behavior [38].
The authors have added the following paper as ref. 38:
- Russo, G.I.; Soeterik, T.; Puche-Sanz, I.; Broggi, G.; Lo Giudice, A.; De
Nunzio, C.; Lombardo, R.; Marra, G.; Gandaglia, G. Prostate Cancer Prostatic
Dis. 2022, in press.

Round 2
Reviewer 2 Report
The authors revised the paper according to my major comments. One issue should be addressed before publication: The paper should be edit by English mother laguage
Reviewer 3 Report
The authors revised the manuscript fully and properly. Congratulations.